# How to Start Training:
# The Effect of Initialization and Architecture

**Boris Hanin**
Department of Mathematics
Texas A& M University
College Station, TX, USA
bhanin@math.tamu.edu

**David Rolnick**
Department of Mathematics
Massachusetts Institute of Technology
Cambridge, MA, USA
drolnick@mit.edu

## Abstract

We identify and study two common failure modes for early training in deep $\mathrm{ReLU}$ nets. For each, we give a rigorous proof of when it occurs and how to avoid it, for fully connected, convolutional, and residual architectures. We show that the first failure mode, exploding or vanishing mean activation length, can be avoided by initializing weights from a symmetric distribution with variance $2$/fan-in and, for ResNets, by correctly scaling the residual modules. We prove that the second failure mode, exponentially large variance of activation length, never occurs in residual nets once the first failure mode is avoided. In contrast, for fully connected nets, we prove that this failure mode can happen and is avoided by keeping constant the sum of the reciprocals of layer widths. We demonstrate empirically the effectiveness of our theoretical results in predicting when networks are able to start training. In particular, we note that many popular initializations fail our criteria, whereas correct initialization and architecture allows much deeper networks to be trained.

## 1 Introduction

Despite the growing number of practical uses for deep learning, training deep neural networks remains a challenge. Among the many possible obstacles to training, it is natural to distinguish two kinds: problems that prevent a given neural network from ever achieving better-than-chance performance and problems that have to do with later stages of training, such as escaping flat regions and saddle points [12, 27], reaching spurious local minima [5, 15], and overfitting [2, 34]. This paper focuses specifically on two failure modes related to the first kind of difficulty:

**(FM1)**: The mean length scale in the final layer increases/decreases exponentially with the depth.

**(FM2)**: The empirical variance of length scales across layers grows exponentially with the depth.

Our main contributions and conclusions are:

- **The mean and variance of activations in a neural network are both important in determining whether training begins.** If both failure modes FM1 and FM2 are avoided, then a deeper network need not take longer to start training than a shallower network.

- **FM1 is dependent on weight initialization.** Initializing weights with the correct variance (in fully connected and convolutional networks) and correctly weighting residual modules (in residual networks) prevents the mean size of activations from becoming exponentially large or small as a function of the depth, allowing training to start for deeper architectures.

- **For fully connected and convolutional networks, FM2 is dependent on architecture.** Wider layers prevent FM2, again allowing training to start for deeper architectures. In the

case of constant-width networks, the width should grow approximately linearly with the depth to avoid FM2.

- **For residual networks, FM2 is largely independent of the architecture.** Provided that residual modules are weighted to avoid FM1, FM2 can never occur. This qualitative difference between fully connected and residual networks can help to explain the empirical success of the latter, allowing deep and relatively narrow networks to be trained more readily.

FM1 for fully connected networks has been previously studied [8, 21, 25]. Training may fail to start, in this failure mode, since the difference between network outputs may exceed machine precision even for moderate $d$. For $\mathrm{ReLU}$ activations, FM1 has been observed to be overcome by initializations of He et al. [8]. We prove this fact rigorously (see Theorems 5 and 6). We find empirically that for poor initializations, training fails more frequently as networks become deeper (see Figures 1 and 4).

Aside from [29], there appears to be less literature studying FM1 for residual networks (ResNets) [9]. We prove that the key to avoiding FM1 in ResNets is to correctly rescale the contributions of individual residual modules (see Theorems 5 and 6). Without this, we find empirically that training fails for deeper ResNets (see Figure 2).

FM2 is more subtle and does not seem to have been widely studied (see [16] for a notable exception). We find that FM2 indeed impedes early training (see Figure 3). Let us mention two possible explanations. First, if the variance between activations at different layers is very large, then some layers may have very large or small activations that exceed machine precision. Second, the backpropagated SGD update for a weight $w$ in a given layer includes a factor that corresponds to the size of activations at the previous layer. A very small update of $w$ essentially keeps it at its randomly initialized value. A very large update on the hand essentially re-randomizes $w$. Thus, we conjecture that FM2 causes the stochasticity of parameter updates to outweigh the effect of the training loss.

Our analysis of FM2 reveals an interesting difference between fully connected and residual networks. Namely, for fully connected and convolutional networks, FM2 is a function of architecture, rather than just of initialization, and can occur even if FM1 does not. For residual networks, we prove by contrast that FM2 never occurs once FM1 is avoided (see Corollary 2 and Theorem 6).

## 2 Related Work

Closely related to this article is the work of Taki [29] on initializing ResNets as well as the work of Yang-Schoenholz [32, 33]. The article [29] gives heuristic computations related to the mean squared activation in a depth-$L$ ResNet and suggests taking the scales $\eta_\ell$ of the residual modules to all be equal to $1/L$ (see §3.2). The articles [32, 33], in contrast, use mean field theory to derive the dependence of both forward and backward dynamics in a randomly initialized ResNet on the residual module weights.

Also related to this work is that of He et al. [8] already mentioned above, as well as [14, 22, 23, 25]. The authors in the latter group show that information can be propagated in infinitely wide $\mathrm{ReLU}$ nets so long as weights are initialized independently according to an appropriately normalized distribution (see condition (ii) in Definition 1). One notable difference between this collection of papers and the present work is that we are concerned with a rigorous computation of finite width effects.

These finite size corrections were also studied by Schoenholz et al. [26], which gives exact formulas for the distribution of pre-activations in the case when the weights and biases are Gaussian. For more on the Gaussian case, we also point the reader to Giryes et al. [6]. The idea that controlling means and variances of activations at various hidden layers in a deep network can help with the start of training was previously considered in Klaumbauer et al. [16]. This work introduced the scaled exponential linear unit (SELU) activation, which is shown to cause the mean values of neuron activations to converge to $0$ and the average squared length to converge to $1$. A different approach to this kind of self-normalizing behavior was suggested in Wu et al. [31]. There, the authors suggest adding a linear hidden layer (that has no learnable parameters) but directly normalizes activations to have mean $0$ and variance $1$. Activation lengths can also be controlled by constraining weight matrices to be orthogonal or unitary (see e.g. [1, 10, 13, 18, 24]).

We would also like to point out a previous appearance in Hanin [7] of the sum of reciprocals of layer widths, which we here show determines the variance of the sizes of the activations (see

Theorem [5]) in randomly initialized fully connected ReLU nets. The article [7] found that this same sum of reciprocals is also related to the problem of vanishing and exploding gradients, indicating commonalities between this failure mode and those we consider here. Finally, we point the reader to the discussion around Figure 6 in [2], which also finds that time to convergence is better for wider networks.

## 3 Results

In this section, we will (1) provide an intuitive motivation and explanation of our mathematical results, (2) verify empirically that our predictions hold, and (3) show by experiment the implications for training neural networks.

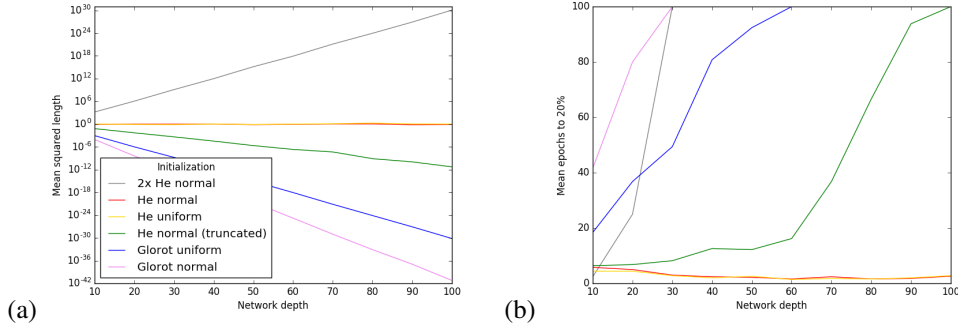

(a)                                                                                          (b)

Figure 1: Comparison of the behavior of differently initialized fully connected networks as depth increases. Width is equal to depth throughout. Note that in `He normal (truncated)`, the normal distribution is truncated at two standard deviations from the mean, as implemented e.g. in Keras and PyTorch. For `2x He normal`, weights are drawn from a normal distribution with twice the variance of `He normal`. (a) Mean square length $M_d$ (log scale), demonstrating exponential decay or explosion unless variance is set at $2$/fan-in, as in `He normal` and `He uniform` initializations; (b) average number of epochs required to obtain 20% test accuracy when training on vectorized MNIST, showing that exponential decay or explosion of $M_d$ is associated with reduced ability to begin training.

### 3.1   Avoiding FM1 for Fully Connected Networks: Variance of Weights

Consider a depth-$d$, fully connected ReLU net $\mathcal{N}$ with hidden layer widths $n_j$, $j = 0, \ldots, d$, and random weights and biases (see Definition [1] for the details of the initialization). As $\mathcal{N}$ propagates an input vector $\mathrm{act}^{(0)} \in \mathbf{R}^{n_0}$ from one layer to the next, the lengths of the resulting vectors of activations $\mathrm{act}^{(j)} \in \mathbf{R}^{n_j}$ change in some manner, eventually producing an output vector whose length is potentially very different from that of the input. These changes in length are summarized by

$$M_j := \frac{1}{n_j} \left\| \mathrm{act}^{(j)} \right\|^2,$$

where here and throughout the squared norm of a vector is the sum of the squares of its entries. We prove in Theorem [5] that the mean of the normalized output length $M_d$, which controls whether failure mode FM1 occurs, is determined by the variance of the distribution used to initialize weights. We emphasize that all our results hold for *any fixed input*, which need not be random; we average only over the weights and the biases. Thus, FM1 cannot be directly solved by batch normalization [11], which renormalizes by averaging over inputs to $\mathcal{N}$, rather than averaging over initializations for $\mathcal{N}$.

**Theorem 1** (FM1 for fully connected networks (informal)). *The mean $\mathbb{E}\left[M_d\right]$ of the normalized output length is equal to the input length if network weights are drawn independently from a symmetric distribution with variance $2$/fan-in. For higher variance, the mean $\mathbb{E}\left[M_d\right]$ grows exponentially in the depth $d$, while for lower variance, it decays exponentially.*

In Figure [1], we compare the effects of different initializations in networks with varying depth, where the width is equal to the depth (this is done to prevent FM2, see §[3.3]). Figure [1](a) shows that, as

predicted, initializations for which the variance of weights is smaller than the critical value of 2/fan-in lead to a dramatic decrease in output length, while variance larger than this value causes the output length to explode. Figure 1(b) compares the ability of differently initialized networks to start training; it shows the average number of epochs required to achieve 20% test accuracy on MNIST [19]. It is clear that those initializations which preserve output length are also those which allow for fast initial training - in fact, we see that it is *faster* to train a suitably initialized depth-100 network than it is to train a depth-10 network. Datapoints in (a) represent the statistics over random unit inputs for 1,000 independently initialized networks, while (b) shows the number of epochs required to achieve 20% accuracy on vectorized MNIST, averaged over 5 training runs with independent initializations, where networks were trained using stochastic gradient descent with a fixed learning rate of 0.01 and batch size of 1024, for up to 100 epochs. Note that changing the learning rate depending on depth could be used to compensate for FM1; choosing the right initialization is equivalent and much simpler.

While $\mathbb{E}[M_d]$ and its connection to the variance of weights at initialization have been previously noted (we refer the reader especially to [8] and to §2 for other references), the implications for choosing a good initialization appear not to have been fully recognized. Many established initializations for ReLU networks draw weights i.i.d. from a distribution for which the variance is not normalized to preserve output lengths. As we shall see, such initializations hamper training of very deep networks. For instance, as implemented in the Keras deep learning Python library [4], the only default initialization to have the critical variance 2/fan-in is `He uniform`. By contrast, `LeCun uniform` and `LeCun normal` have variance 1/fan-in, `Glorot uniform` (also known as `Xavier uniform`) and `Glorot normal` (`Xavier normal`) have variance 2/(fan-in + fan-out). Finally, the initialization `He normal` comes close to having the correct variance, but, at the time of writing, the Keras implementation truncates the normal distribution at two standard deviations from the mean (the implementation in PyTorch [20] does likewise). This leads to a decrease in the variance of the resulting weight distribution, causing a catastrophic decay in the lengths of output activations (see Figure 1). We note this to emphasize both the sensitivity of initialization and the popularity of initializers that can lead to FM1.

It is worth noting that the 2 in our optimal variance 2/fan-in arises from the ReLU, which zeros out symmetrically distributed input with probability $1/2$, thereby effectively halving the variance at each layer. (For linear activations, the 2 would disappear.) The initializations described above may preserve output lengths for activation functions *other than ReLU*. However, ReLU is one of the most common activation functions for feed-forward networks and various initializations are commonly used blindly with ReLUs without recognizing the effect upon ease of training. An interesting systematic approach to predicting the correct multiplicative constant in the variance of weights as a function of the non-linearity is proposed in [22, 25] (e.g., the definition of $\chi_1$ around (7) in Poole et al. [22]). For non-linearities other than $\mathrm{ReLU}$, however, this constant seems difficult to compute directly.

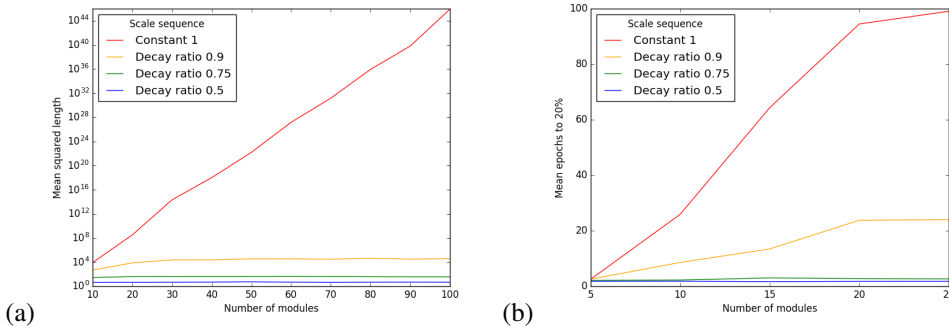

Figure 2: Comparison of the behavior of differently scaled ResNets as the number of modules increases. Each residual module here is a single layer of width 5. (a) Mean length scale $M_L^{res}$, which grows exponentially in the sum of scales $\eta_\ell$; (b) average number of epochs to 20% test accuracy when training on MNIST, showing that $M_L^{res}$ is a good predictor of initial training performance.

## 3.2 Avoiding FM1 for Residual Networks: Weights of Residual Modules

To state our results about FM1 for ResNets, we must set some notation (based on the framework presented e.g. in Veit et al. [30]). For a sequence $\eta_\ell$, $\ell = 1, 2 \ldots$ of positive real numbers and a sequence of fully connected ReLU nets $\mathcal{N}_1, \mathcal{N}_2, \ldots$, we define a *residual network* $\mathcal{N}_L^{res}$ with *residual modules* $\mathcal{N}_1, \ldots, \mathcal{N}_L$ and *scales* $\eta_1, \ldots, \eta_L$ by the recursion

$$\mathcal{N}_0^{res}(x) = x, \qquad \mathcal{N}_\ell^{res}(x) = \mathcal{N}_{\ell-1}^{res}(x) + \eta_\ell \mathcal{N}_\ell \left( \mathcal{N}_{\ell-1}^{res}(x) \right), \qquad \ell = 1, \ldots, L.$$

Explicitly,

$$\begin{aligned} \mathcal{N}_L^{res}(x) \;=\; & x \;+\; \eta_1 \mathcal{N}_1(x) \;+\; \eta_2 \mathcal{N}_2(x + \eta_1 \mathcal{N}_1(x)) \\ & +\; \eta_3 \mathcal{N}_3 \left( x + \eta_1 \mathcal{N}_1(x) + \eta_2 \mathcal{N}_2 \left( x + \eta_1 \mathcal{N}_1(x) \right) \right) + \cdots . \end{aligned} \tag{1}$$

Intuitively, the scale $\eta_\ell$ controls the size of the correction to $\mathcal{N}_{\ell-1}^{res}$ computed by the residual module $\mathcal{N}_\ell$. Since we implicitly assume that the depths and widths of the residual modules $\mathcal{N}_\ell$ are uniformly bounded (e.g., the modules may have a common architecture), failure mode FM1 comes down to determining for which sequences $\{\eta_\ell\}$ of scales there exist $c, C > 0$ so that

$$c \leq \sup_{L \geq 1} \mathbb{E}\left[ M_L^{res} \right] \leq C, \tag{2}$$

where we write $M_L^{res} = \| \mathcal{N}_L^{res}(x) \|^2$ and $x$ is a unit norm input to $\mathcal{N}_L^{res}$. The expectation in (2) is over the weights and biases in the fully connected residual modules $\mathcal{N}_\ell$, which we initialize as in Definition 1, except that we set biases to zero for simplicity (this does not affect the results below). A part of our main theoretical result, Theorem 6, on residual networks can be summarized as follows.

**Theorem 2** (FM1 for ResNets (informal)). *Consider a randomly initialized ResNet with $L$ residual modules, scales $\eta_1, \ldots, \eta_L$, and weights drawn independently from a symmetric distribution with variance $2/\text{fan-in}$. The mean $\mathbb{E}\left[ M_L^{res} \right]$ of the normalized output length grows exponentially with the sum of the scales $\sum_{\ell=1}^{L} \eta_\ell$.*

We empirically verify the predictive power of the quantity $\eta_\ell$ in the performance of ResNets. In Figure 2(a), we initialize ResNets with constant $\eta_\ell = 1$ as well as geometrically decaying $\eta_\ell = b^\ell$ for $b = 0.9, 0.75, 0.5$. All modules are single hidden layers with width 5. We observe that, as predicted by Theorem 1, $\eta_\ell = 1$ leads to exponentially growing length scale $M_L^{res}$, while $\eta_\ell = b^\ell$ leads the mean of $M_L^{res}$ to grow until it reaches a plateau (the value of which depends on $b$), since $\sum \eta_\ell$ is finite. In Figure 2(b), we show that the mean of $M_L^{res}$ well predicts the ease with which ResNets of different depths are trained. Note the large gap between $b = 0.9$ and $0.75$, which is explained by noting that the approximation of $\eta^2 \ll \eta$ which we use in the proof holds for $\eta \ll 1$, leading to a larger constant multiple of $\sum \eta_\ell$ in the exponent for $b$ closer to 1. Each datapoint is averaged over 100 training runs with independent initializations, with training parameters as in Figure 1.

## 3.3 FM2 for Fully Connected Networks: The Effect of Architecture

In the notation of §3.1, failure mode FM2 is characterized by a large expected value for

$$\widehat{\text{Var}}[M] := \frac{1}{d} \sum_{j=1}^{d} M_j^2 - \left( \frac{1}{d} \sum_{j=1}^{d} M_j \right)^2,$$

the empirical variance of the normalized squared lengths of activations among all the hidden layers in $\mathcal{N}$. Our main theoretical result about FM2 for fully connected networks is the following.

**Theorem 3** (FM2 for fully connected networks (informal)). *The mean $\mathbb{E}[\widehat{\text{Var}}[M]]$ of the empirical variance for the lengths of activations in a fully connected ReLU net is exponential in $\sum_{j=1}^{d-1} 1/n_j$, the sum of the reciprocals of the widths of the hidden layers.*

For a formal statement see Theorem 5. It is well known that deep but narrow networks are hard to train, and this result provides theoretical justification; since for such nets $\sum 1/n_j$ is large. More than that, this sum of reciprocals gives a definite way to quantify the effect of "deep but narrow" architectures on the volatility of the scale of activations at various layers within the network. We note that this result also implies that for a given depth and fixed budget of neurons or parameters, constant

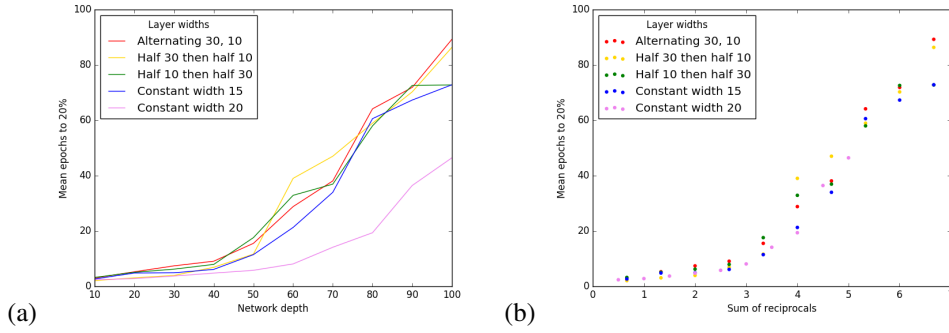

(a)  (b)

Figure 3: Comparison of ease with which different fully connected architectures may be trained. (a) Mean epochs required to obtain 20% test accuracy when training on MNIST, as a function of network depth; (b) same $y$-axis, with $x$-axis showing the sum of reciprocals of layer widths. Training efficiency is shown to be predicted closely by this sum of reciprocals, independent of other details of network architecture. Note that all networks are initialized with `He normal` weights to avoid FM1.

width is optimal, since by the Power Mean Inequality, $\sum_j 1/n_j$ is minimized for all $n_j$ equal if $\sum n_j$ (number of neurons) or $\sum n_j^2$ (approximate number of parameters) is held fixed.

We experimentally verify that the sum of reciprocals of layer widths (FM2) is an astonishingly good predictor of the speed of early training. Figure 3(a) compares training performance on MNIST for fully connected networks of five types:

    i. Alternating layers of width 30 and 10,

    ii. The first half of the layers of width 30, then the second half of width 10,

    iii. The first half of the layers of width 10, then the second half of width 30,

    iv. Constant width 15,

    v. Constant width 20.

Note that types (i)-(iii) have the same layer widths, but differently permuted. As predicted by our theory, the order of layer widths does not affect FM2. We emphasize that

$$\frac{1}{30} + \frac{1}{10} = \frac{1}{15} + \frac{1}{15},$$

type (iv) networks have, for each depth, the same sum of reciprocals of layer widths as types (i)-(iii). As predicted, early training dynamics for networks of type (i)-(iv) were similar for each fixed depth. By contrast, networks of type (v), which had a lower sum of reciprocals of layer widths, trained faster for every depth than the corresponding networks of types (i)-(iv).

In all cases, training becomes harder with greater depth, since $\sum 1/n_j$ increases with depth for constant-width networks. In Figure 3(b), we plot the same data with $\sum 1/n_j$ on the $x$-axis, showing this quantity's power in predicting the effectiveness of early training, irrespective of the particular details of the network architecture in question.

Each datapoint is averaged over 100 independently initialized training runs, with training parameters as in Figure 1. All networks are initialized with `He normal` weights to prevent FM1.

### 3.4  FM2 for Residual Networks

In the notation of §3.2, failure mode FM2 is equivalent to a large expected value for the empirical variance

$$\widehat{\mathrm{Var}}[M^{res}] := \frac{1}{L} \sum_{\ell=1}^{L} (M_\ell^{res})^2 - \left( \frac{1}{L} \sum_{\ell=1}^{L} M_\ell^{res} \right)^2$$

of the normalized squared lengths of activations among the residual modules in $\mathcal{N}$. Our main theoretical result about FM2 for ResNets is the following (see Theorem 5 for the precise statement).

**Theorem 4** (FM2 for ResNets (informal)). *The mean $\mathbb{E}[\widehat{\mathrm{Var}}[M^{res}]]$ of the empirical variance for the lengths of activations in a residual* ReLU *net with L residual modules and scales $\eta_\ell$ is exponential in $\sum_{\ell=1}^{L}\eta_\ell$. By Theorem 2, this means that in ResNets, if failure mode FM1 does not occur, then neither does FM2 (assuming FM2 does not occur in individual residual modules).*

## 3.5  Convolutional Architectures

Our above results were stated for fully connected networks, but the logic of our proofs carries over to other architectures. In particular, similar statements hold for convolutional neural networks (ConvNets). Note that the fan-in for a convolutional layer is not given by the width of the preceding layer, but instead is equal to the number of features multiplied by the kernel size.

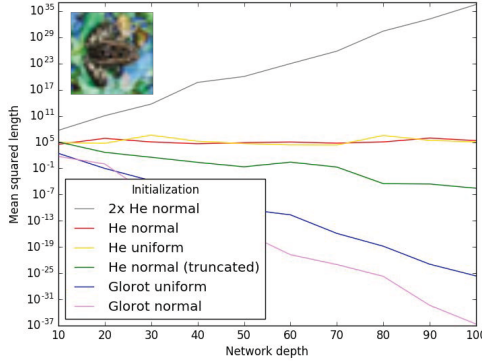

Figure 4: Comparison of the behavior of differently initialized ConvNets as depth increases, with the number of features at each layer proportional to the overall network depth. The mean output length over different random initializations is observed to follow the same patterns as in Figure 1 for fully connected networks. Weight distributions with variance 2/fan-in preserve output length, while other distributions lead to exponential growth or decay. The input image from CIFAR-10 is shown.

In Figure 4, we show that the output length behavior we observed in fully connected networks also holds in ConvNets. Namely, mean output length equals input length for weights drawn i.i.d. from a symmetric distribution of variance 2/fan-in, while other variances lead to exploding or vanishing output lengths as the depth increases. In our experiments, networks were purely convolutional, with no pooling or fully connected layers. By analogy to Figure 1, the fan-in was set to approximately the depth of the network by fixing kernel size $3 \times 3$ and setting the number of features at each layer to one tenth of the network's total depth. For each datapoint, the network was allowed to vary over 1,000 independent initializations, with input a fixed image from the dataset CIFAR-10 [17].

## 4  Notation

To state our results formally, we first give the precise definition of the networks we study; and we introduce some notation. For every $d \geq 1$ and $\mathbf{n} = (n_i)_{i=0}^{d} \in \mathbf{Z}_+^{d+1}$, we define

$$\mathfrak{N}(\mathbf{n}, d) = \left\{ \begin{array}{c} \text{fully connected feed-forward nets with ReLU activations} \\ \text{and depth } d, \text{ whose } j^{th} \text{ hidden layer has width } n_j \end{array} \right\}.$$

Note that $n_0$ is the dimension of the input. Given $\mathcal{N} \in \mathfrak{N}(\mathbf{n}, d)$, the function $f_\mathcal{N}$ it computes is determined by its weights and biases

$$\{w_{\alpha,\beta}^{(j)}, b_\beta^{(j)}, \ 1 \leq \alpha \leq n_j, \ 1 \leq \beta \leq n_{j+1}, \ 0 \leq j \leq d-1\}.$$

For every input $\mathrm{act}^{(0)} = \left(\mathrm{act}_\alpha^{(0)}\right)_{\alpha=1}^{n_0} \in \mathbf{R}^{n_0}$ to $\mathcal{N}$, we write for all $j = 1, \ldots, d$

$$\mathrm{preact}_\beta^{(j)} = b_\beta^{(j)} + \sum_{\alpha=1}^{n_{j-1}} \mathrm{act}_\alpha^{(j-1)} w_{\alpha,\beta}^{(j)}, \qquad \mathrm{act}_\beta^{(j)} = \mathrm{ReLU}(\mathrm{preact}_\beta^{(j)}), \quad 1 \leq \beta \leq n_j. \quad (3)$$

The vectors $\mathrm{preact}^{(j)}$, $\mathrm{act}^{(j)}$ are thus the inputs and outputs of nonlinearities in the $j^{th}$ layer of $\mathcal{N}$.

**Definition 1** (Random Nets). *Fix $d \geq 1$, positive integers $\mathbf{n} = (n_0, \ldots, n_d) \in \mathbf{Z}_+^{d+1}$, and two collections of probability measures $\mu = \left(\mu^{(1)}, \ldots, \mu^{(d)}\right)$ and $\nu = \left(\nu^{(1)}, \ldots, \nu^{(d)}\right)$ on $\mathbf{R}$ such that $\mu^{(j)}, \nu^{(j)}$ are symmetric around 0 for every $1 \leq j \leq d$, and such that the variance of $\mu^{(j)}$ is $2/(n_{j-1})$.*

*A random network $\mathcal{N} \in \mathfrak{N}_{\mu,\nu}(\mathbf{n}, d)$ is obtained by requiring that the weights and biases for neurons at layer $j$ are drawn independently from $\mu^{(j)}, \nu^{(j)}$, respectively.*

# 5 Formal statements

We begin by stating our results about fully connected networks. Given a random network $\mathcal{N} \in \mathfrak{N}_{\mu,\nu}(d, \mathbf{n})$ and an input $\mathrm{act}^{(0)}$ to $\mathcal{N}$, we write as in §3.1, $M_j$ for the normalized square length of activations $\frac{1}{n_d} \| \mathrm{act}^{(j)} \|^2$ at layer $j$. Our first theoretical result, Theorem 5, concerns both the mean and variance of $M_d$. To state it, we denote for any probability measure $\lambda$ on $\mathbf{R}$ its moments by

$$\lambda_k := \int_{\mathbf{R}} x^k d\lambda(x).$$

**Theorem 5.** *For each $j \geq 0$, fix $n_j \in \mathbf{Z}_+$. For each $d \geq 1$, let $\mathcal{N} \in \mathfrak{N}_{\mu,\nu}(d, \mathbf{n})$ be a fully connected ReLU net with depth $d$, hidden layer widths $\mathbf{n} = (n_j)_{j=0}^d$ as well as random weights and biases as in Definition 1. Fix also an input $\mathrm{act}^{(0)} \in \mathbf{R}^{n_0}$ to $\mathcal{N}$ with $\| \mathrm{act}^{(0)} \| = 1$. We have almost surely*

$$\limsup_{d\to\infty} M_d < \infty \iff \sum_{j\geq 1} \nu_2^{(j)} < \infty. \tag{4}$$

*Moreover, if (4) holds, then there exists a random variable $M_\infty$ (that is almost surely finite) such that $M_d \to M_\infty$ as $d \to \infty$ pointwise almost surely. Further, suppose $\mu_4^{(j)} < \infty$ for all $j \geq 1$ and that $\sum_{j=1}^\infty \left( \nu_2^{(j)} \right)^2 < \infty$. Then*

$$\exp\left[ \frac{1}{2} \sum_{j=1}^d \frac{1}{n_j} \right] \;\leq\; \mathbb{E}\left[ M_d^2 \right] \;\leq\; C_1 \exp\left[ C_2 \sum_{j=1}^d \frac{1}{n_j} \right], \tag{5}$$

*where $C_1, C_2$ the following finite constants:*

$$C_1 = 1 + M_0^2 + \left( M_0 + \sum_{j=1}^\infty \nu_2^{(j)} \right) \sum_{j=1}^\infty \nu_2^{(j)}$$

*and*

$$C_2 = \sup_{d\geq 1} \max\{1, \frac{1}{4}\nu_4^{(d)} - \frac{1}{2}\left( \nu_2^{(d)} \right)^2, 2|\widetilde{\mu}_4^{(d)} - 3|\} \left( 2 + M_0 + \sum_{j=1}^\infty \nu_2^{(j)} \right).$$

*If $M_0 = 1$ and $\sum_{j=1}^\infty \nu_2^{(j)}, \nu_4^{(d)} \leq 1$ and $\mu^{(j)}$ is Gaussian for all $j \geq 1$, then we can take*

$$C_1 = C_2 = 4.$$

*In particular, $\mathrm{Var}[M_d]$ is exponential in $\sum_{j=1}^d 1/n_j$ and if $\sum_j 1/n_j < \infty$, then the convergence of $M_d$ to $M_\infty$ is in $L^2$ and $\mathrm{Var}[M_\infty] < \infty$.*

The proof of Theorem 5 is deferred to the Supplementary Material. Although we state our results only for fully connected feed-forward ReLU nets, the proof techniques carry over essentially verbatim to any feed-forward network in which only weights in the same hidden layer are tied. In particular, our results apply to convolutional networks in which the kernel sizes are uniformly bounded. In this case, the constants in Theorem 5 depend on the bound for the kernel dimensions, and $n_j$ denotes the fan-in for neurons in the $(j+1)^{st}$ hidden layer (i.e. the number of channels in layer $j$ multiplied by the size of the appropriate kernel). We also point out the following corollary, which follows immediately from the proof of Theorem 5.

**Corollary 1** (FM1 for Fully Connected Networks)**.** *With notation as in Theorem 5, suppose that for all $j = 1, \ldots, d$, the weights in layer $j$ of $\mathcal{N}_d$ have variance $\kappa \cdot 2/n_j$ for some $\kappa > 0$. Then the average squared size $M_d$ of activations at layer $d$ will grow or decay exponentially unless $\kappa = 1$:*

$$\mathbb{E}\left[ \frac{1}{n_d} \left\| \mathrm{act}^{(d)} \right\|^2 \right] = \frac{\kappa^d}{n_0} \left\| \mathrm{act}^{(0)} \right\|^2 + \sum_{j=1}^d \kappa^{d-j} \nu_2^{(d)}.$$

Our final result about fully connected networks is a corollary of Theorem 5, which explains precisely when failure mode FM2 occurs (see §3.3). It is proved in the Supplementary Material.

**Corollary 2** (FM2 for Fully Connected Networks). *Take the same notation as in Theorem 5. There exist $c, C > 0$ so that*

$$\frac{c}{d} \sum_{j=1}^{d} \frac{j-1}{d} \exp\left( c \sum_{k=j}^{d-1} \frac{1}{n_k} \right) \leq \mathbb{E}\left[ \widehat{\mathrm{Var}}[M] \right] \leq \frac{C}{d} \sum_{j=1}^{d} \frac{j-1}{d} \exp\left( C \sum_{k=j}^{d-1} \frac{1}{n_k} \right). \tag{6}$$

*In particular, suppose the hidden layer widths are all equal, $n_j = n$. Then, $\mathbb{E}[\widehat{\mathrm{Var}}[M]]$ is exponential in $\beta = \sum_j 1/n_j = (d-1)/n$ in the sense that there exist $c, C > 0$ so that*

$$c \exp\left( c\beta \right) \leq \mathbb{E}\left[ \widehat{\mathrm{Var}}[M] \right] \leq C \exp\left( C\beta \right). \tag{7}$$

Finally, our main result about residual networks is the following:

**Theorem 6.** *Take the notation from §3.2 and §3.4. The mean of squared activations in $\mathcal{N}_L^{res}$ are uniformly bounded in the number of modules $L$ if and only if the scales $\eta_\ell$ form a convergent series:*

$$0 < \sup_{L \geq 1, \, \|x\|=1} M_L^{res} < \infty \qquad \Longleftrightarrow \qquad \sum_{\ell=1}^{\infty} \eta_\ell < \infty. \tag{8}$$

*Moreover, for any sequence of scales $\eta_\ell$ for which $\sup_\ell \eta_\ell < 1$ and for every $K, L \geq 1$, we have*

$$\mathbb{E}\left[ (M_L^{res})^K \right] = \exp\left( O\left( \sum_{\ell=1}^{L} \eta_\ell \right) \right),$$

*where the implied constant depends on $K$ but not on $L$. Hence, once the condition in part (8) holds, both the moments $\mathbb{E}\left[ (M_L^{res})^K \right]$ and the mean of the empirical variance of $\{ M_\ell^{res}, \ \ell = 1, \ldots, L \}$ are uniformly bounded in $L$.*

# 6   Conclusion

In this article, we give a rigorous analysis of the layerwise length scales in fully connected, convolutional, and residual ReLU networks at initialization. We find that a careful choice of initial weights is needed for well-behaved mean length scales. For fully connected and convolutional networks, this entails a critical variance for i.i.d. weights, while for residual nets this entails appropriately rescaling the residual modules. For fully connected nets, we prove that to control not merely the mean but also the variance of layerwise length scales requires choosing a sufficiently wide architecture, while for residual nets nothing further is required. We also demonstrate empirically that both the mean and variance of length scales are strong predictors of early training dynamics.

Our results hold for any fixed input, representing a reasonable (and tractable) approximation to distributions of interest in which inputs are drawn from discrete clusters. In the case of classification problems, for example, high output variance for a single fixed input could indicate high variance over the set of similar inputs of that class. Theoretical analysis over an entire specified input distribution is tricky (and likely depends on the particular distribution in question), though in future work we will attempt to extend the present results to the case of joint distributions over two or more inputs. We also plan to extend our analysis to other (e.g. sigmoidal) activations, recurrent networks, and weight initializations beyond i.i.d., such as orthogonal weights.

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
