[Supplementary Material]

# A   Proof of Theorem 5

Let us first verify that $M_d$ is a submartingale for the filtration $\{\mathcal{F}_d\}_{d\geq 1}$ with $\mathcal{F}_d$ being the sigma algebra generated by all weights and biases up to and including layer $d$ (for background on sigma algebras and martingales we refer the reader to Chapters 2 and 37 in [3]). Since $\text{act}^{(0)}$ is a fixed non-random vector, it is clear that $M_d$ is measurable with respect to $\mathcal{F}_d$. We have

$$
\begin{aligned}
\mathbb{E}\left[M_d \mid \mathcal{F}_{d-1}\right] &= \frac{1}{n_d}\mathbb{E}\left[\left\|\text{act}^{(d)}\right\|^2 \mid \text{act}^{(d-1)}\right] \\
&= \frac{1}{n_d}\sum_{\beta=1}^{n_d}\mathbb{E}\left[\left(\text{preact}_\beta^{(d)}\right)^2 \mathbf{1}_{\{\text{preact}_\beta^{(d)}>0\}} \mid \text{act}^{(d-1)}\right],
\end{aligned}
\tag{9}
$$

where we can replace the sigma algebra $\mathcal{F}_{d-1}$ by the sigma algebra generated by $\text{act}^{(d-1)}$ since the computation done by a feed-forward neural net is a Markov chain with respect to activations at consecutive layers (for background see Chapter 8 in [3]). Next, recall that by assumption the weights and biases are symmetric in law around $0$. Note that for each $\beta$, changing the signs of all the weights $w_{\alpha,\beta}^{(d)}$ and biases $b_\beta^{(d)}$ causes $\text{preact}_\beta^{(d)}$ to change sign. Hence, we find

$$
\mathbb{E}\left[\left(\text{preact}_\beta^{(d)}\right)^2 \mathbf{1}_{\{\text{preact}_\beta^{(d)}>0\}} \mid \text{act}^{(d-1)}\right] = \mathbb{E}\left[\left(\text{preact}_\beta^{(d)}\right)^2 \mathbf{1}_{\{\text{preact}_\beta^{(d)}<0\}} \mid \text{act}^{(d-1)}\right].
$$

Note that

$$
\text{preact}_\beta^{(d)}\left(\mathbf{1}_{\{\text{preact}_\beta^{(d)}>0\}} + \mathbf{1}_{\{\text{preact}_\beta^{(d)}<0\}}\right) = \text{preact}_\beta^{(d)}.
$$

Symmetrizing the expression in (9), we obtain

$$
\begin{aligned}
\mathbb{E}\left[M_d \mid \mathcal{F}_{d-1}\right] &= \frac{1}{2n_d}\sum_{\beta=1}^{n_d}\mathbb{E}\left[\left(\text{preact}_\beta^{(d)}\right)^2 \mid \text{act}^{(d-1)}\right] \\
&= \frac{1}{2n_d}\sum_{\beta=1}^{n_d}\mathbb{E}\left[\left(b_\beta^{(d)} + \sum_{\alpha=1}^{n_{d-1}}\text{act}_\alpha^{(d-1)}\,w_{\alpha,\beta}^{(d)}\right)^2 \mid \text{act}^{(d-1)}\right] \\
&= \frac{1}{2}\nu_2^{(d)} + \frac{1}{n_{d-1}}\left\|\text{act}^{(d-1)}\right\|^2 \geq M_{d-1},
\end{aligned}
\tag{10}
$$

where in the second equality we used that the weights $w_{\alpha,\beta}^{(d)}$ and biases $b_\beta^{(d)}$ are independent of $\mathcal{F}_{d-1}$ with mean $0$ and in the last equality that $\text{Var}[w_{\alpha,\beta}^{(d)}] = 2/n_{j-1}$. The above computation also yields that for each $d \geq 1$,

$$
\mathbb{E}[M_d] = \mathbb{E}\left[\frac{1}{n_d}\left\|\text{act}^{(d)}\right\|^2\right] = \frac{1}{n_0}\left\|\text{act}^{(0)}\right\|^2 + \frac{1}{2}\sum_{j=1}^{d}\nu_2^{(j)}.
\tag{11}
$$

It also shows that $\widehat{M}_d = M_d - \sum_{j=1}^{d}\frac{1}{2}\nu_2^{(d)}$ is a martingale. Taking the limit $d \to \infty$ in (11) proves (4). Next, assuming condition (4), we find that

$$
\sup_{d\geq 1}\mathbb{E}\left[\max\{M_d,0\}\right] \leq \frac{1}{n_0}\left\|\text{act}^{(0)}\right\|^2 + \frac{1}{2}\sum_{j=1}^{\infty}\nu_2^{(j)},
$$

which is finite. Hence, we may apply Doob's pointwise martingale convergence theorem (see Chapter 35 in [3]) to conclude that the limit

$$
M_\infty = \lim_{d\to\infty}M_d
$$

is exists and is finite almost surely. Indeed, Doob's result states that if our martingale $\widehat{M}_d$ is bounded in $L^1$ uniformly in $d$, then, almost surely, $\widehat{M}_d$ has a finite pointwise limit as $d \to \infty$. To show (5) we will need the following result.

**Lemma 1.** *Fix $d \geq 1$. Then*

$$\frac{M_{d-1}^2 + (6 - \nu_2^{(d)})M_{d-1}}{n_d} \leq \mathrm{Var}[M_d \mid \mathcal{F}_{d-1}] \leq \frac{C(1 + M_{d-1}^2 + M_{d-1})}{n_d},$$

*where*

$$C = \max\{1, \tfrac{1}{2}\mu_4^{(d)} - \tfrac{1}{4}\left(\mu_2^{(d)}\right)^2, 2|\widetilde{\mu}_4^{(d)} - 3|\}.$$

*Proof.* Note that

$$M_d = \frac{1}{n_d} \sum_\beta \left(\mathrm{act}_\beta^{(d)}\right)^2,$$

and, conditioned on $\mathrm{act}^{(d-1)}$, the random variables $\{\mathrm{act}_\beta^{(d)}\}_\beta$ are i.i.d. Hence,

$$\mathrm{Var}[M_d \mid \mathcal{F}_{d-1}] = \frac{1}{n_d} \mathrm{Var}\left[\left(\mathrm{act}_1^{(d)}\right)^2 \mid \mathcal{F}_{d-1}\right]. \tag{12}$$

We apply the same symmetrization trick as in the derivation of (10) to obtain

$$\mathbb{E}\left[\left(\mathrm{act}_1^{(d)}\right)^4 \mid \mathcal{F}_{d-1}\right] = \frac{1}{2}\mathbb{E}\left[\left(\mathrm{preact}_1^{(d)}\right)^4 \mid \mathrm{act}^{(d-1)}\right]$$

$$= \frac{1}{2}\mathbb{E}\left[\left(\sum_{\alpha=1}^{n_{d-1}} \mathrm{act}_\alpha^{(d-1)} w_{\alpha,1}^{(d)} + b_1^{(d)}\right)^4 \mid \mathrm{act}^{(d-1)}\right],$$

which after using that the odd moments of $w_{\alpha,1}^{(d)}$ and $b_1^{(d)}$ vanish becomes

$$\frac{1}{2}\mathbb{E}\left[\left(\sum_{\alpha=1}^{n_{d-1}} \mathrm{act}_\alpha^{(d-1)} w_{\alpha,1}^{(d)}\right)^4 \mid \mathrm{act}^{(d-1)}\right] + \frac{6\nu_2^{(d)}}{n_{d-1}} \left\|\mathrm{act}^{(d-1)}\right\|^2 + \frac{1}{2}\nu_4^{(d)}.$$

To evaluate the first term, note that

$$\mathbb{E}\left[\left(\sum_{\alpha=1}^{n_{d-1}} \mathrm{act}_\alpha^{(d-1)} w_{\alpha,1}^{(d)}\right)^4 \mid \mathrm{act}^{(d-1)}\right] = \sum_{\substack{\alpha_i=1 \\ 1 \leq i \leq 4}}^{n_{d-1}} \prod_{i=1}^{4} \mathrm{act}_{\alpha_i}^{(d-1)} \mathbb{E}\left[\prod_{i=1}^{4} w_{\alpha_i,1}^{(d)}\right].$$

Since

$$\mathbb{E}\left[\prod_{i=1}^{4} w_{\alpha_i,1}^{(d)}\right] = \frac{4}{n_{d-1}^2}\left[\mathbf{1}_{\{\substack{\alpha_1=\alpha_2 \\ \alpha_3=\alpha_4}\}} + \mathbf{1}_{\{\substack{\alpha_1=\alpha_3 \\ \alpha_2=\alpha_4}\}} + \mathbf{1}_{\{\substack{\alpha_1=\alpha_4 \\ \alpha_2=\alpha_3}\}} + (\widetilde{\mu}_4^{(d)} - 3)\mathbf{1}_{\{\alpha_1=\alpha_2=\alpha_3=\alpha_4\}}\right],$$

we conclude that

$$\mathbb{E}\left[\left(\sum_{\alpha=1}^{n_{d-1}} \mathrm{act}_\alpha^{(d-1)} w_{\alpha,1}^{(d)}\right)^4 \mid \mathrm{act}^{(d-1)}\right] = \frac{4}{n_{d-1}^2}\left(3\left\|\mathrm{act}^{(d-1)}\right\|^4 + \left(\widetilde{\mu}_4^{(d)} - 3\right)\left\|\mathrm{act}^{(d-1)}\right\|_4^4\right),$$

where we recall that $\widetilde{\mu}_4^{(d)} = \mu_4^{(d)} / \left(\mu_2^{(d)}\right)^2$. Putting together the preceding computations and using that

$$\mathbb{E}\left[\left(\mathrm{act}_\beta^{(d)}\right)^2 \mid \mathrm{act}^{(d-1)}\right] = M_{d-1} + \frac{1}{2}\nu_2^{(d)},$$

we find that

$$\mathrm{Var}\left[\left(\mathrm{act}_1^{(d)}\right)^2 \mid \mathcal{F}_{d-1}\right] = 5M_{d-1}^2 + \frac{2(\widetilde{\mu}_4^{(d)} - 3)}{n_{d-1}^2}\left\|\mathrm{act}^{(d-1)}\right\|_4^4$$

$$+ \left(6 - \nu_2^{(d)}\right)M_{d-1} + \frac{1}{2}\nu_4^{(d)} - \frac{1}{4}\left(\nu_2^{(d)}\right)^2.$$

Recall that the excess kurtosis $\widetilde{\mu}_4^{(d)} - 3$ of $\mu^{(d)}$ is bounded below by $-2$ for any probability measure (see Chapter 4 in [28]) and observe that $\| \operatorname{act}^{(d-1)} \|_4^4 \leq \| \operatorname{act}^{(d-1)} \|^4$. Therefore, using that $\frac{1}{2}\nu_4^{(d)} - \frac{1}{4}(\nu_2^{(d)})^2 \geq 0$, we obtain

$$M_{d-1}^2 + (6 - \nu_2^{(d)})M_{d-1} \leq \operatorname{Var}\left[ \left(\operatorname{act}_1^{(d)}\right)^2 \mid \mathcal{F}_{d-1}\right] \leq C(1 + M_{d-1}^2 + M_{d-1})$$

with

$$C = \max\{1, \frac{1}{2}\nu_4^{(d)} - \frac{1}{2}\left(\mu_2^{(d)}\right)^2, 2\,|\widetilde{\mu}_4| - 3\}.$$

This completes the proof of the Lemma. $\qquad\square$

To conclude the proof of Theorem 5, we write

$$\mathbb{E}\left[M_d^2 \mid \mathcal{F}_{d-1}\right] = \operatorname{Var}[M_d \mid \mathcal{F}_{d-1}] + \left(M_{d-1} + \frac{1}{2}\nu_2^{(d)}\right)^2$$

and combine Lemma 1 with the expression (11) to obtain with $C$ as in Lemma 1

$$\mathbb{E}\left[M_d^2 \mid \mathcal{F}_{d-1}\right] \leq \frac{C}{n_d}\left(1 + M_{d-1}^2 + M_d\right) + M_{d-1}^2 + M_{d-1}\nu_2^{(d)} + \frac{1}{4}\nu_4^{(d)}$$

and

$$\mathbb{E}\left[M_d^2 \mid \mathcal{F}_{d-1}\right] \geq M_{d-1}^2\left(1 + \frac{1}{n_d}\right) + M_d\left(\nu_2^{(d)} + \frac{6 - \nu_2^{(d)}}{n_d}\right) \geq M_{d-1}^2\left(1 + \frac{1}{n_d}\right).$$

Taking expectations of both sides in the inequalities above yields with $C$ as in Lemma 1

$$\mathbb{E}\left[M_{d-1}^2\right]\left(1 + \frac{1}{n_d}\right) \leq \mathbb{E}\left[M_d^2\right] \leq \left(a_d + \mathbb{E}\left[M_{d-1}^2\right]\right)\left(1 + \frac{C}{n_d}\right),$$

where

$$a_d \leq \frac{C}{n_d}\left(1 + M_0 + \sum_{j=1}^{\infty}\nu_2^{(j)}\right) + \nu_2^{(d)}\left(M_0 + \sum_{j=1}^{\infty}\nu_2^{(j)}\right) + \left(\nu_2^{(d)}\right)^2$$

and therefore

$$\sum_{j=1}^{d}a_d \leq C\left(1 + M_0 + \sum_{j=1}^{\infty}\nu_2^{(j)}\right)\sum_{j=1}^{d}\frac{1}{n_j} + \left(M_0 + \sum_{j=1}^{\infty}\nu_2^{(j)}\right)\sum_{j=1}^{d}\nu_2^{(j)} + \sum_{j=1}^{d}\left(\nu_2^{(j)}\right)^2$$

$$\leq \left(1 + M_0^2 + \left(M_0 + \sum_{j=1}^{\infty}\nu_2^{(j)}\right)\sum_{j=1}^{d}\nu_2^{(j)} + \sum_{j=1}^{d}\left(\nu_2^{(j)}\right)^2\right)\left(1 + C\left(1 + M_0 + \sum_{j=1}^{\infty}\nu_2^{(j)}\right)\sum_{j=1}^{d}\frac{1}{n_j}\right).$$

Iterating the lower bound in this inequality yields the lower bound in (5). Similarly, using that $1 + C/n_d > 1$, we iterate the upper bound to obtain

$$\mathbb{E}\left[M_d^2\right] \leq \left(a_d + \mathbb{E}\left[M_{d-1}^2\right]\right)\left(1 + \frac{C}{n_d}\right)$$

$$\leq \left(a_d + a_{d-1} + \mathbb{E}\left[M_{d-2}^2\right]\right)\left(1 + \frac{C}{n_d}\right)\left(1 + \frac{C}{n_{d-1}}\right)$$

$$\cdots \leq \left(\sum_{j=1}^{d}a_j + M_0^2\right)\exp\left(C\sum_{j=1}^{d}\frac{1}{n_j}\right).$$

Using the above estimate for $\sum_j a_j$, gives the upper bound in (5) and completes the proof of Theorem 5.

# B    Proof of Corollary 2

Fix a fully connected ReLU net $\mathcal{N}$ with depth $d$ and hidden layer widths $n_0, \ldots, n_d$. We fix an input $\text{act}^{(0)}$ to $\mathcal{N}$ and study the empirical variance $\widehat{\text{Var}}[M]$ of the squared sizes of activations $M_j$, $j = 1, \ldots, d$. Since the biases in $\mathcal{N}$ are 0, the squared activations $M_j$ are a martingale (see (10)) and we find

$$\mathbb{E}\left[M_j M_{j'}\right] = \mathbb{E}\left[M_{\min\{j,j'\}}^2\right].$$

Thus, using that by (5) for some $c > 0$

$$\mathbb{E}\left[M_j^2\right] \geq c \exp\left(c \sum_{k=j}^{d-1} \frac{1}{n_k}\right),$$

we find

$$\mathbb{E}\left[\widehat{\text{Var}}_d\right] = \frac{1}{d} \sum_{j=1}^{d} \mathbb{E}\left[M_j^2\right] - \frac{1}{d^2} \sum_{j,j'=1}^{d} \mathbb{E}\left[M_j M_{j'}\right]$$

$$= \frac{1}{d} \sum_{j=1}^{d} \mathbb{E}\left[M_j^2\right] - \frac{1}{d^2} \sum_{j=1}^{d} (d - j + 1)\mathbb{E}\left[M_j^2\right]$$

$$\geq \frac{1}{d} \sum_{j=1}^{d} \frac{j-1}{d} c \exp\left(c \sum_{k=j}^{d-1} \frac{1}{n_k}\right).$$

To see that this sum is exponential in $\sum 1/n_j$ as in (7), let us consider the special case of equal widths $n_j = n$. Then, writing

$$\beta = \sum_{j=1}^{d-1} \frac{1}{n_j},$$

we have

$$\sum_{j=1}^{d} \frac{j-1}{d} c \exp\left(c \sum_{k=j}^{d-1} \frac{1}{n_k}\right) \approx c \int_0^1 x e^{\beta c(1-x)} dx = \frac{e^{\beta c}}{\beta^2 c} + \frac{1}{\beta}\left(1 - \frac{1}{\beta c}\right).$$

This proves the lower bounds in (6) and (7). The upper bounds are similar.

# C    Proof of Theorem 6

To understand the sizes of activations produced by $\mathcal{N}_L^{res}$, we need the following Lemma.

**Lemma 2.** *Let $\mathcal{N}$ be a feed-forward, fully connected* ReLU *net with depth $d$ and hidden layer widths $n_0, \ldots, n_d$ having random weights as in Definition 1 and biases set to 0. Then for each $\eta \in (0,1)$, we have*

$$\|x + \eta \mathcal{N}(x)\|^2 = \|x\|^2 (1 + O(\eta)).$$

*Proof of Lemma.* We have:

$$\mathbb{E}\left[\|x + \eta \mathcal{N}(x)\|\right]^2 = \mathbb{E}\left[\|x\|^2 + 2\eta \langle x, \mathcal{N}(x)\rangle + \eta^2 \|\mathcal{N}(x)\|^2\right] \tag{13}$$

$$= \|x\|^2 \left(1 + \eta^2 + 2\eta\mathbb{E}\left[\langle \widehat{x}, \mathcal{N}(\widehat{x})\rangle\right]\right),$$

where $\widehat{x} = \frac{x}{\|x\|}$, and we have used the fact that $\|\mathcal{N}(x)\|^2 = \|x\|^2$ (see (10)) as well as the positive homogeneity of ReLU nets with zero biases:

$$\mathcal{N}(\lambda x) = \lambda \mathcal{N}(x), \qquad \lambda > 0.$$

Write

$$\mathbb{E}\left[\langle x, \mathcal{N}(x)\rangle\right] = \sum_{\beta=1}^{n} x_\beta \mathbb{E}\left[\mathcal{N}_\beta(x)\right].$$

Let us also write $x = x^{(0)}$ for the input to $\mathcal{N}$, similarly set $x^{(j)}$ for the activations at layer $j$. We denote by $W_\beta^{(j)}$ the $\beta^{th}$ row of the weights $W^{(j)}$ at layer $j$ in $\mathcal{N}$. We have:

$$
\begin{aligned}
\mathbb{E}\left[\mathcal{N}_\beta(x) \mid x^{(d-1)}\right] &= \mathbb{E}\left[x_\beta^{(d)}\right] = \mathbb{E}\left[W_\beta^{(d)} x^{(d-1)} \mathbf{1}_{\{W_\beta^{(d-1)} x\} > 0} \mid x^{(d-1)}\right] \\
&= \mathbb{E}\left[\left|W_\beta^{(d)} x^{(d-1)}\right| \mathbf{1}_{\{W_\beta^{(d)} x^{(d-1)} > 0\}} \mid x^{(d-1)}\right] \\
&= \frac{1}{2} \mathbb{E}\left[\left|W_\beta^{(d)} x^{(d-1)}\right| \mid x^{(d-1)}\right] \\
&\leq \frac{1}{2}\left(\mathbb{E}\left[\left|W_\beta^{(d)} x^{(d-1)}\right|^2 \mid x^{(d-1)}\right]\right)^{1/2} \\
&= \frac{1}{\sqrt{2n}}\left\|x^{(d-1)}\right\|.
\end{aligned}
$$

Therefore, using that $\left\|x^{(j)}\right\|$ is a supermartingale (since its square is a martingale by (10)):

$$
\mathbb{E}\left[\mathcal{N}_\beta(x)\right] \leq \frac{1}{\sqrt{2n}} \mathbb{E}\left[\left\|x^{(d-1)}\right\|\right] \leq \frac{1}{\sqrt{2n}}\left\|x^{(0)}\right\| = \frac{1}{\sqrt{2n}}\|x\|.
$$

Hence, we obtain:

$$
\mathbb{E}\left[\langle \widehat{x}, \mathcal{N}(\widehat{x})\rangle\right] = O\left(\frac{\|\widehat{x}\|_1}{\sqrt{n}}\right) = O(1)
$$

since by Jensen's inequality,

$$
\sum_{j=1}^n |x_i| \leq \sqrt{n} \sum_{j=1}^n x_i^2, \qquad x_i \in \mathbb{R}.
$$

Combining this with (13) completes the proof. $\qquad\square$

The Lemma implies part (i) of the Theorem as follows:

$$
\begin{aligned}
\mathbb{E}\left[M_L^{res}\right] &= \mathbb{E}\left[\left\|\mathcal{N}_{L-1}^{res}(x) + \eta_L \mathcal{N}_L\left(\mathcal{N}_{L-1}^{res}(x)\right)\right\|^2\right] \\
&= \mathbb{E}\left[\mathbb{E}\left[\left\|\mathcal{N}_{L-1}^{res}(x) + \eta_L \mathcal{N}_L\left(\mathcal{N}_{L-1}^{res}(x)\right)\right\|^2 \mid \mathcal{N}_{L-1}^{res}(x)\right]\right] \\
&= (1 + O(\eta_L)) \mathbb{E}\left[M_{L-1}^{res}\right],
\end{aligned}
$$

where we used the fact that $\eta_\ell^2 = O(\eta_\ell)$ since $\eta_\ell \in (0,1)$. Iterating this inequality yields

$$
\mathbb{E}\left[M_L^{res}\right] = \prod_{\ell=1}^L (1 + O(\eta_\ell)) = \exp\left(\sum_{\ell=1}^L \log(1 + O(\eta_\ell))\right) = \exp\left(O\left(\sum_{\ell=1}^L \eta_\ell\right)\right),
$$

Derivation of the estimates (ii) follows exactly the same procedure and hence is omitted. Finally, using these estimates, we find that the mean empirical variance of $\{M_\ell^{res}\}$ is exponential in $\sum_\ell \eta_\ell$:

$$
\begin{aligned}
\mathbb{E}\left[\frac{1}{L}\sum_{\ell=1}^L (M_\ell^{res})^2 - \left(\frac{1}{L}\sum_{\ell=1}^L M_\ell^{res}\right)^2\right] &\leq \frac{1}{L}\sum_{\ell=1}^L \mathbb{E}\left[(M_\ell^{res})^2\right] \\
&= \frac{1}{L}\sum_{\ell=1}^L \exp\left(O\left(\sum_{j=1}^\ell \eta_j\right)\right) \\
&= \exp\left(O\left(\sum_{j=1}^L \eta_j\right)\right).
\end{aligned}
$$