[Reviews · NeurIPS 2018]

Reviewer 1



The paper studies two failure modes during the initial phase of deep network training: 1. The mean length of activation vector in the last layer increases/decreases exponentially with depth. The mean is taken over the randomness in weights/biases while the input is considered fixed. 2. The variance of length of activation vector grows exponentially with depth. It has been shown that FM1 in fully connected architectures can be avoided by sampling the weights of the network from a symmetric distribution with variance 2/fan-in. For ResNets, it has been shown that in addition to weight initialization, each residual block needs to be scaled appropriately such that the sum of reciprocal of these scaling factors is close to 1 (the latter is not mentioned explicitly/clearly). It is shown that FM2 is avoided for fully connected networks when using large width in networks. More specifically, the variance in the length of activation vector is exponential in the sum of reciprocal of widths of layers. For ResNets it is shown that addressing FM1 automatically rules out FM2. All theoretical results are complemented side-by-side with appropriate experiments. For FM1, the mean length of activations is shown with depth for different initializations along with the number of epochs needed to achieve 20% performance. For FM2, only the latter is shown (It would be informative to also show how the empirical variance behaves for different cases). Comments: FM1 is well studied in literature for fully connection architecture as acknowledged by authors as well. The novelty seems to be twofold: 1. FM1 has not been well studied for ResNet architecture. It is studied in this paper. 2. When studying FM1, previous literature assumes the input is random and the variance of activations is calculated over both the randomness in input and parameter initialization distribution. In this paper, input is considered fixed. Regarding point 2 above, I am not sure I understand why preserving the mean length of activation vector in the last layer for a *fixed input* is important. In previous work, the motivation is to preserve the information content in the input. As a proxy, this information is measured by the variance in input and so calculating the variance of hidden units over the input/parameter distribution makes sense. The reasoning for considering fixed input is not discussed in the paper. Please clarify. Also, since the theory predicts that FM2 in fully connected networks can be avoided by using network widths such that the sum of reciprocal is small, I am curious if it equivalent to use something like widths 10,20,30,40 vs widths 40,30,20,10 in a network. In these two cases, the sum of reciprocal of widths is identical, but one has width increasing with depth while the other has width decreasing with depth. My intuition is that the latter may have a poor "mean epochs to 20%". I recommend the authors to add this experiment. If the results are unexpected, it would be interesting to leave it as an open thread, and in the other case it would make the theory more compelling. The authors should acknowledge in line 173 that the paper "A Closer Look at Memorization in Deep Networks" discussed (their Figure 6) that time to convergence is better for wider networks. (I did not go through the proofs.) Clarity: The novelty of the paper is not stated clearly. For instance, the first two contributions mentioned in the paper are already known. The novelty in these contributions is not clear. There are two problems in the way the authors have introduced FM1 and FM2: 1. Clearly identifying FM1 and FM2 in previous work and better motivating why they (can) cause failure is very important and does not seem to be done properly. Instead, after stating FM1 and FM2 in the introduction, the rest of the paper discusses how to solve it. This is especially the case with FM2 which is not well known. 2. While it is stated in the definition of FM1 and FM2 that quantities increase/decrease exponentially with depth, and that it is problematic, it is never stated explicitly how these quantities should ideally behave: mean length should be preserved and variance should be as small as possible. Originality: As mentioned above, the novelty needs to be stated clearly and the motivation for FM2 and why FM1 is considered for fixed input needs to be clarified. Overall, I think the paper is somewhat original. The drawback is that initializing networks parameters from distributions with variance 2/fan-in is already known. But the part about ResNet is novel. The part about why large width is important is also novel and very interesting. Significance: I think the intialization scheme for ResNet (scaling residual blocks seems to be the novel part) can be of importance for practitioners. I have given an overall score of 6 mainly because of lack of clarity in the paper. ## Post rebuttal update: The authors clarified some of the questions I raised. Although the experiment on training a network with decreasing width is not conducted for the rebuttal as I proposed, the authors agree to include it in the final submission. However I have two other concerns: 1. The paper talks about preserving activation length and variance which was originally proposed as a way to preserve information during the forward pass of a deep network. However there is no discussion on gradient norm preservation which is an important for avoiding vanishing/exploding gradients problem during initialization and is also important for avoiding the initial failure during training. 2. The constant C that appears in theorem 5 is only discussed to the point that it exists and is positive. But no further discussion is provided. This is a concern because a very large C satisfying theorem 5 would be trivial since the activation length will effectively not be preserved. This is further important because the theorem talks about any fixed input vector. When considering a large number of input vectors (a dataset), it is more likely for the variance in activation norm is large for some of the samples. For these reasons, I am sticking with my original score.

Reviewer 2



Summary This paper covers a theoretical basis for training deep neural networks and how initialization/architecture choices can solve early training problems. This paper focuses on early training "failure modes" that prevent a network from just getting off the ground ("better-than-chance" performance), rather than improving performance at later stages. The two early stage failure modes they wish to avoid are: 1. mean length scale in the final layer increases/decreases exponentially with the depth. 2. empirical variance of length scales across layers grows exponentially with the depth. The paper fits in with literature and recent interest in ReLU activation functions and residual networks to ease training so one can construct deeper neural nets. Their results are: - For fully connected nets, there is a critical variance to use for initialization distribution to avoid failure #1 - For fully connected nets, wider and constant width layers helps avoid failure #2 - For Residual networks, decaying weights on the residual modules help avoid *both* #1 and #2 Quality The paper's motivations are apparent and strong, providing a rigorous basis for the variance of the distribution for the weight initialization and subsequent architectural considerations on residual nets. Particularly of interest is the fact that both failure modes can be prevented by the same process of minimizing the sum of the weightings on the residual modules. This leads to very practical considerations, however -- for a network of a given depth, it just indicates that your weightings on modules should exponentially decay, or have some sort of convergent sequence. For a fully connected net of given depth, the width of each layer should be maximized and equal widths are ideal -- to minimize the sum of reciprocals of layer depth -- to avoid the second failure mode. These are practical but perhaps simple / incremental ideas. Clarity Although I appreciate the attempt to motivate results for the reader, and cover the same material with increasing rigor, sections 1, 3, and 4 could be reorganized in perhaps a more efficient manner, to avoid a sense of some redundancy of sentences. I think at the end of line 90 you should add the words "the normalized output length", as you refer to it in as such a sentence later, but I think just including this quick naming before the formula could only increase clarity on a first read. Figure 1 is a good motivating figure, adds to understanding beyond the text. Figure 3 is a less convincing figure. Originality The fact that many popular initializations fail to meet the criteria presented in the paper (like Xavier, truncated He, etc.) shows that the knowledge in the paper is not immensely widespread prior to its publication. The paper does rely heavily on reference [8], He ResNet paper -- initialization with critical variance is already implemented in PyTorch. Although only one of the possible, popular initializes meets their criteria -- so the results of this paper would help you choose the correct initializes among current options, not choose a new one. Perhaps the greatest argument against the paper could be that the results are only incremental on the ResNet and ReLU results in the literature. Avoiding failure mode 1 for residual nets seems like a possibly obvious result / conclusion -- you need a bounded sequence for a bounding on the expected length scale. Significance A good theme for this paper comes around Line 114, roughly paraphrased by me here: other workarounds to solve this problem exist, but our suggestion is equivalent and much simpler (and possibly addresses multiple problems in one). The work applies to deep nets with ReLU activation functions. The authors note how the work might be extended to other functions, or more rigorously to convolutional architectures, but those are further directions. Of course including more architectures or activations adds more to the results, but it may especially be warranted as some of the papers results were perhaps forshadowed by earlier work. The papers strength is that the results are well motivated, intuitively presented, and rigorous. I could certainly see the work being extended to more architectures, activations, and failure modes in early training of deep nets. However, the ability to easily extend could be seen as a weakness as well -- perhaps there were obvious further directions to take the work that were left unexplored. As the paper itself built upon an variance for initialization previously discovered, perhaps the results did not move the needle enough. (post-rebuttal) The proposed change to figure 3 as well as the additional text should help the reader. I would insist on this being done.

Reviewer 3



The paper proposes an analysis of the behavior of the activities in a neural network, according to the initialization of the weights and the architecture. It addresses two problems: (1) the activities explode or vanish with the depth and (2) their variance grows exponentially with the depth.The article provides conditions over the initialization of the weights and the architecture of the neural network in order to avoid these failures. Experiments and a theoretical analysis prove that these conditions are sufficient in the cases of fully connected neural networks and residual networks. The article is part of the field of weight initialization and analysis of the activities, such as "Resurrecting the sigmoid" (2017, Pennington and Schoenholz). The main contribution of this article lies in the generality of the theorems: no limitation over the number of layers, no assumption about the data set. Moreover, the sketch of their proof is likely to be reusable. The only strong restriction is the use of ReLU as activation function. The experiments in the case of fully connected networks show that the problems (1) and (2) can occur when the conditions given by the article are not fulfilled. The proof of theorem 5, that gives sufficient conditions to avoid problem (1), is clear and rigorous. Nevertheless, for the problem (2), it misses experiments with neural networks with non-decreasing layer width (*) (for example in figures 3a and 3b). These are necessary to evaluate the condition given to avoid problem (2): this condition (sum of reciprocal of layer widths) is possibly far from necessary. Concerning proofs: (a) The proof of corollary 2 is not clear: on line 400, the approximation is either incomplete or false (the constant c disappears, and exp(\beta x) should be exp(\beta (1 - x))). However, replacing the inequality on line 396 by the first inequality of equation 5 leads exactly to the claimed result on line 400. (**) (b) Theorem 5: the claim that limsup(M_d) is finite (equation 4) does not appear to be proven. The fact that E[M_d] is uniformly bounded is not sufficient. (***) (c) typos in theorem 5 and its proof: line 236: a constant N is defined but is never used; the second-order moment \nu_2^{(d)} is frequently squared when it should not. In general, the paper is very well written, the main ideas are clearly explained and arise readily. Overall, this paper provides a good analysis of the problem of exploding/vanishing activations and gives an interesting point of view to study activations. The results over their variance are a good start, but suffer from a lack of experiments (see (*)). Some parts of the proofs need improvement (see (**) and (***)). If the last two points are solved, this article should be published.